# Cytomembrane Trafficking Pathways of Connexin 26, 30, and 43

**DOI:** 10.3390/ijms241210349

**Published:** 2023-06-19

**Authors:** Yan-Jun Zong, Xiao-Zhou Liu, Lei Tu, Yu Sun

**Affiliations:** 1Department of Otorhinolaryngology, Union Hospital, Tongji Medical College, Huazhong University of Science and Technology, Wuhan 430022, China; u201810264@hust.edu.cn (Y.-J.Z.); d202181808@hust.edu.cn (X.-Z.L.); 2Division of Gastroenterology, Union Hospital, Tongji Medical College, Huazhong University of Science and Technology, Wuhan 430022, China; 3Hubei Province Key Laboratory of Oral and Maxillofacial Development and Regeneration, Wuhan 430022, China

**Keywords:** connexin 43, connexin 30, connexin 26, gap junctions, transport

## Abstract

The connexin gene family is the most prevalent gene that contributes to hearing loss. Connexins 26 and 30, encoded by *GJB2* and *GJB6*, respectively, are the most abundantly expressed connexins in the inner ear. Connexin 43, which is encoded by *GJA1*, appears to be widely expressed in various organs, including the heart, skin, the brain, and the inner ear. The mutations that arise in *GJB2*, *GJB6*, and *GJA1* can all result in comprehensive or non-comprehensive genetic deafness in newborns. As it is predicted that connexins include at least 20 isoforms in humans, the biosynthesis, structural composition, and degradation of connexins must be precisely regulated so that the gap junctions can properly operate. Certain mutations result in connexins possessing a faulty subcellular localization, failing to transport to the cell membrane and preventing gap junction formation, ultimately leading to connexin dysfunction and hearing loss. In this review, we provide a discussion of the transport models for connexin 43, connexins 30 and 26, mutations affecting trafficking pathways of these connexins, the existing controversies in the trafficking pathways of connexins, and the molecules involved in connexin trafficking and their functions. This review can contribute to a new way of understanding the etiological principles of connexin mutations and finding therapeutic strategies for hereditary deafness.

## 1. Introduction

Gap junctions are sites of cell-to-cell contact that allow the bidirectional transit of intercellular cytoplasmic ions, second messengers, and other substances with molecular weights less than 1.5 kDa (e.g., Ca^2+^, K^+^, and inositol triphosphates (IP_3_)) [1,2]. In mammals, each gap junction consists of two half-channels, called connexons. Although a gap is left between adjacent cell membranes, the two connexons interact and dock in the extracellular space to form a tightly sealed intercellular hydrophilic pore [3]. Each connexon is composed of six tetraspan membrane protein subunits called connexins (Cxs). Connexins are a multi-gene family and the proteins range in size from 25 to 62 kDa, with a common membrane topology, including intracellular amino (NT) and carboxyl termini (CT), four transmembrane domains, a cytoplasmic loop (CL), and two highly conserved extracellular loops [4,5]. Connexins are predicted to include at least 20 isoforms in humans with widely varying amino acid sequences in their intracellular CT structural domains [6]. The CT and its post-translational modifications play an important role in Cx biology by regulating their ubiquitination and assembly into higher-level structures [7,8].

In mammals, the organ of Corti is a complex neuroepithelium formed by a combination of mechanosensory hair cells that perceive sounds and supporting cells. Within the organ of Corti, sensory inner and outer hair cells, as well as non-sensory supporting cells, are arranged in a regular mosaic pattern that extends along the cochlear duct from the base to the apices. Connexin 26 (Cx26) and Connexin 30 (Cx30) are common Cxs in the cochlea and are mainly expressed in the support cells of the organ of Corti, the basal and intermediate cells of the stria vascularis, and fibroblasts in the spiral ligament [9,10,11,12,13]. In the cochlea, Cx26 and Cx30 assemble into two types of gap junctions that form a syncytium that extends from the spiral limbus to the spiral ligament. Epidemiological studies have shown that, despite strong genetic heterogeneity, up to 50% of cases of autosomal recessive non-comprehensive deafness are associated with mutations in *GJB2*, which encodes Cx26 and is involved in homeostasis in the inner ear [14,15]. *GJB2* mutations also cause comprehensive deafness associated with dermatological diseases of several different prognoses [16,17]. Mutations in *GJB6*, which encodes Cx30, have also been associated with the development of comprehensive or non-comprehensive hereditary hearing loss [18]. Cx26 plays an important role in the formation of gap junction plaque (GJP) in the cochlea and the maintenance of normal hearing. Yum et al. [19] demonstrated that, in Hela cells, *GJB2* mutation has a trans-dominant-negative effect on Cx30, which may be due to excessive endocytosis caused by Cx26 deficiency in the inner sulcus cells [20], resulting in a dramatic reduction of Cx30 GJPs. Similarly, animal experiments have shown that overexpression of Cx26 in Cx30^−/−^ mice can compensate for the functional deficit of Cx30, restore auditory sensitivity, and prevent hair cell death [21]. However, the overexpression of Cx30 in Cx26^−/−^ mice did not lead to an opening of the Corti tunnel or rescue the severe hearing loss [22]. More seriously, in Cx26 R75W, a dominant-negative mutant of Cx26, even the overexpression of wild-type Cx26 failed to normalize the length of the GJPs in mutant mice [20]. Kamiva et al. [20] also demonstrated that the formation of normal GJPs is dependent on the presence of Cx26 using a chimera model. Since neighboring cells express different types of Cxs, different contact surfaces of the same cell form different GJPs; that is, only when both cells express Cx26 can they form large, functional GJPs at the contact surface. However, if one cell is deficient in Cx26, even if Cx30 is abundantly expressed, only fragmented vesicle-like GJPs composed of Cx30 can be formed. Cx43 is the most prevalent connexin and plays key roles in the heart, skin, and brain [23]. In the heart, Cx43 gap junctions are concentrated in the intercalated disc that connects the two ends of cardiomyocytes [24], which is where action potential propagation is ensured. Altered Cx43 gap junction distribution after myocardial ischemia can lead to malignant ischemic arrhythmias [25]. Cx43 is also widely expressed in the inner ear, including the supporting cells, spiral ligament, stria vascularis, and Schwann and satellite cells of spiral ganglion neurons [26]. Cx43 is recorded in the cochlear bone, which encases the cochlea, and in cells located in the auditory brainstem and midbrain relays known as the cochlear nucleus, the eighth cranial nerve, the lateral lemniscus, the olivary complex, and the inferior colliculus, which are in charge of the transmission of electrical signals around the brain [26]. Furthermore, mutations in *GJA1*, encoding Cx43, can also contribute to hearing loss.

Because of the many isoforms of Cxs, the biosynthesis, structural composition, and degradation of Cxs must be precisely regulated for gap junctions to properly function. Here, we review the transport models for Cx43, which has a long CT, as well as Cx30 and Cx26, which have a short CT. We also discuss existing controversies and the molecules involved in Cxs transport and their functions. This review may help formulate new ideas for unraveling the pathogenic principles of Cx mutations and the selection of better treatment options.

## 2. Cxs Mutations Induced Dysfunction and Abnormal Trafficking Pathways

Studies have shown that there are six types of Cxs mutations associated with deafness in human patients [27]. Class I mutations are those that prevent the formation of gap junctions. Class II includes mutations that do not affect the formation of gap junctions, but the mutated gap junctions display ineffective function. Class III refers to mutations that specifically impair gap junction-mediated biochemical coupling. Class IV consists of mutations that result in functional gain owing to abnormal hemichannel opening. No reported mutations with effects on gap junction function possibly representing polymorphisms are included in Class V. Class VI covers mutations that have not been thoroughly studied in vitro. In particular, Class I is due to Cxs mutations resulting in dysfunction at various stages of the connexins life cycle [28], including protein synthesis, trafficking/direction to the plasma membrane, membrane insertion and assembly into connexins, and degradation. Mutations that cause Cxs to fail to form gap junctions based on in vitro studies are shown in Table 1.

Variants belonging to Class I may result in altered cell membrane or subcellular localization sequences, ultimately resulting in the failure of the gap junction formation. Prospectively, Class I mutations may also affect the binding of Cxs to other intracellular partners which generally interfaced with connexin subunit [27]. The correct transport pathway is essential for the subcellular localization of connexins and the formation of gap junction channels.

The material exchange and information exchange between adjacent cells are important functions of connexin. Part of the mutations listed in Table 1 have been confirmed to affect the permeability changes of cells to ions or other molecules in vitro [29,50,61,65,66,73]. Therefore, the abnormal traffic pathways and subcellular localization of mutated connexin may be important for the pathological processes of diseases such as hearing loss [26,30,79,80,81,82].

## 3. Possible Transport Pathways and Controversies Regarding Cxs

### 3.1. Transport Pathways of Cx43

The literature suggests that the biological occurrence of Cx43 GJPs is a two-step mechanism; that is, it depends on both microtubules and actin, but there are still some controversies regarding the roles of actin and the sites that fuse to the membrane.

It has been suggested that actin is directly involved in the Cx43 transport process [83,84,85]. The hemichannels synthesized in the trans-Golgi are first transported along microtubules to the perinexus region, a peripheral cell membrane containing unconnected hemichannels around the GJPs. Then, due to tethering [86,87], vesicles aggregate at the end of microtubules and fuse with the plasma membrane of the perinexus region one after another. Finally, the hemichannels interact with cortex actin via zonula occludens (ZO) proteins [23,88], which regulate the directional transport of hemichannels from the perinexus region to the GJPs (Figure 1). It has also been suggested that trans-Golgi-synthesized hemichannels are directly transported along microtubules to GJPs and that the role of actin is mainly to guide the localization of adhesion junctions (AJs) with the assistance of Rho-GTPase [89,90] (Figure 1), thereby indirectly influencing the formation of GJPs rather than being directly involved in hemichannel transport [91]. Evidence in support of the first transport model is twofold. First, Cx43-containing vesicles can fuse with the plasma membrane of isolated cells [92]. Second, the presence of Cx43 was detected on the non-connected surfaces of cells and the membrane mobility of Cx43 could be calculated [83]. Four pieces of evidence supporting the second transport model are as follows. First, there is at least one associated microtubule present in the region containing GJPs [91]. Second, microtubules captured by end-binding protein 1 (EB1), which regulates the growth characteristics of the positive end of microtubules, can reach the region containing GJPs more frequently and stably [91]. Third, after complete photobleaching, GJPs rapidly recover and are consistent with the pre-bleaching plaque morphology rather than appearing around the original plaque first [91]. Fourth, the recovery of GJPs after photobleaching is slowed down when microtubule depolymerization is induced using nocodazole or inhibited using paclitaxel [91], further indicating that the rapid transport of Cx43 is dependent on microtubule-mediated intracellular transport.

In terms of the site that fuses to the membrane, Lauf et al. [83] suggested that Cx43 transport vesicles randomly fuse with the plasma membrane of non-GJPs and then directionally move to form GJPs, since no transport of hemichannels in any specific direction after leaving the Golgi apparatus was observed and hemichannels transported to the plasma membrane of non-GJPs could be observed to freely move (Figure 1). This is consistent with electron microscopy images that indicate the presence of intermembrane particles dispersed around GJPs [93,94], which may act as a reserve pool of Cxs to complement GJPs. It is also consistent with the presence of hemichannels in the plasma membrane, which exist as independent units that regulate the intra- and extracellular environment [95,96,97]. However, Shaw et al. [91] suggested that Cx43 is directly targeted to GJPs along a shorter pre-fusion pathway and that AJs are the preferential transport site for Cx43, as shown in Figure 1. Most of the Cx43 located on the plasma membrane does not move and, after complete photobleaching of GJPs, fluorescence recovers from the center of the plaque without movement from the edge to the center [91].

### 3.2. Transport Pathways of Cx30

For the transport of Cx30, as illustrated in Figure 2, Cx30 hemichannels depend on the Golgi apparatus and are transported through microtubules to the non-GJP region of the cell membrane, and then Cx30 hemichannels interact with actin, which regulates hemichannel transport from the non-GJP region to the preexisting surrounding GJPs [91,98,99,100]. Other studies have suggested that the actin network is closely associated with the formation of Cx30 GJPs and that its role is mainly to facilitate peripheral Cx30 transport from the perinexus regions to GJPs and less in maintaining the stability of GJPs [98,99]. Laser confocal microscopy of mouse cochlear sections confirmed the colocalization of Cx30 GJPs with actin but with slightly different distribution patterns; that is, there was more Cx30 and less actin at GJPs, while there was less Cx30 and more actin at the perinexus region [99]. An in situ proximity ligation assay (PLA) to detect protein interactions in situ showed that Cx30 interacts with β-actin and has a weaker signal at the GJPs than in the perinexus region [99].

However, there is controversy regarding the Cx30 transport network. As to the effect of the Golgi apparatus on Cx30 transport, Cx30 GJPs are not present on the cell membrane of the mutSar1 cell line, in which the mutation inhibits coat protein II (COP II) budding from the endoplasmic reticulum (ER), and thus inhibits classical ER-Golgi transport and Cx30 accumulates in the ER [100] in contrast to normal cells. Kelly et al. [100] treated Hela cells with BFA to depolymerize the Golgi apparatus and showed no reduction in Cx30 at the cell membrane, but Cx30 accumulated in the ER, indicating that Cx30 can resist removal from the cell membrane but cannot leave the ER via the ER-Golgi pathway after disruption of the Golgi apparatus. In contrast, Qu et al. [98] observed no significant changes in Cx30 on the cell member after treating Hela cells with BFA for 5 h, indicating that Cx30 transport is not dependent on the Golgi apparatus.

To assess the impact of microtubules on Cx30 transport, Qu et al. [98] demonstrated that membrane Cx30 was reduced by 23% after microtubule depolymerization. These data indicate that there are two different but simultaneous pathways for Cx30 transport; that is, microtubule-dependent long-distance transport and microtubule-independent short-distance transport via actin or direct interaction between the ER and the plasma membrane [98]. In contrast, Kelly et al. [100] found no significant changes in Cx30 on the cell membrane 10 h after microtubule depolymerization, indicating that Cx30 transport is independent of microtubules.

Regarding the influence of actin on Cx30 transport, both Defourny et al. [99] and Qu et al. [98] showed a dose-dependent reduction of Cx30 in the cell membrane after disrupting the actin network with cytochalasin B. The distribution pattern of Cx30 GJPs gradually changes from linear gap junctions at the edge of the cell membrane to circular or ring-shaped gap junctions that are diffusely distributed throughout the cell with increasing dose. Double immunofluorescence staining showed that Cx30 is colocalized with and distributed along the actin filaments [98]. Together, these results confirm that there is an association between the actin network and the transport of Cx30. However, Kelly et al. [100] showed no significant change in the distribution of Cx30 across the cell membrane 10 h after the disruption of the actin network, indicating that Cx30 transport is not dependent on the actin network.

There is also considerable controversy over whether the sites of fusion of Cx30 hemichannels with membranes are located at the edge of GJPs or GJPs. A study by Kelly et al. [100] showed that after photobleaching, Cx30 GJPs recover from the edge, which means that Cx30 transport vesicles fuse with the plasma membrane at the edge of GJPs. However, Shaw et al. [91] showed the opposite, indicating that the site of Cx30 fusion with the membrane is in the center of the GJPs. Furthermore, PLA experiments confirmed that the interaction of Cx30 with F-actin is asymmetric and that Cx30 is preferentially recruited to one side of the GJPs by actin, indicating that the membrane transport of Cx30 is directional rather than random [91].

### 3.3. Transport Pathways of Cx26

In a number of papers, Defourny et al. [99,101,102,103] postulated the transport pathway of Cx26 by fixation staining of cochlear basement membrane cultures. N-cadherin promotes microtubules to anchor to lipid rafts that are thought to be present at tricellular junctions, and Cx26/30 oligomers are transported along microtubules to the cell surface of lipid raft-rich regions. Then, due to the low affinity of Cx26 for cholesterol, the Cx26/30 hemichannels targeted to the lipid rafts rapidly laterally diffuse to accumulate in the periphery of preexisting GJPs, so there is no functional gap junction at the lipid rafts. Finally, each hemichannel docks with the adjacent cellular hemichannel to form a complete gap junction channel (Figure 3). It has also been proposed that the transport of Cx26 is a one-step mechanism that is distinct from that of long-CT connexins [99,101] and only dependent on microtubules. However, the above model may have some problems that remain to be solved. 

The first issue is the relationship between the position of lipid rafts and GJPs. Lipid rafts are cholesterol-sphingolipid-rich domains that provide a platform for intracellular transport and signal transduction [104]. In the model of Defourny et al. [102], it was observed that Cx26/30 GJPs are located between adjacent cells and lipid rafts are located at the three-cell junctions, with no colocalization between the two, so it was assumed that Cx26 is first targeted to lipid rafts via microtubules and then laterally diffuses to the periphery of GJPs. However, Kamiva et al. [20] showed that lipid rafts exist in the interstitial space between GJPs and that GJPs and lipid rafts are arranged during the interphase between two cells. By observing transfected 293T cells, Schubert et al. [105] determined the colocalization of Cx26 with caveolin-1, which is thought to be a specific type of lipid raft [106].

Secondly, is Cx26 transport actually actin-dependent? Several studies have suggested that actin may not be involved in Cx26 transport. No significant changes in Cx26 GJPs were observed after disruption of the actin network using cytochalasin D [101], suggesting that Cx26 transport is not actin-dependent. Laser confocal microscopy of mice cochlear basement membrane cultures revealed no colocalization of Cx26, which is located between two adjacent cells, with actin, which is situated at the triple cell junctions [98,101]. Detourny et al.’s study [101] also showed no colocalization of Cx26 with ZO-1/2, which is consistent with the conclusion that Cx26 has no PDZ-binding motif [107]. However, several studies have shown that Cx26 transport is dependent on actin. Martin et al.’s study [6] showed that disruption of actin decreased dye transfer via Cx26-GFP by 40% and F-actin was reduced by 54.85% after Cx26 knockdown [108]. In summary, it is still highly controversial whether disruption of the actin network affects Cx26 transport. 

Third, whether the Golgi apparatus and microtubules are involved in the transport of Cx26 is still highly controversial. Some studies have suggested that Cx26 transport is Golgi-dependent, but also microtubule-independent. Thomas et al.’s study [109] revealed that after disruption of the Golgi structure using BFA, Cx26 localized in the reticular structures and failed to translocate to the cell membrane, and after the removal of BFA, Cx26 re-localized to the cell membrane in NRK cells. The use of Sar1 mutant cell lines also confirmed that when the assembly and outgrowth process of COP II nucleocapsid proteins was inhibited, Cx26 was retained within the ER and could not traffic to the cell membrane [109]. However, Cx26 can still be regenerated on the cell membrane surface after microtubule depolymerization [109]. Nevertheless, it has also been suggested that Cx26 can bypass the Golgi apparatus and be transported to the cell membrane via microtubules in a vesicular transport-type manner. The processes of vesicular transport and translocation to the cell membrane to form GJPs of Cx26 are not greatly affected by the disruption of the Golgi apparatus using BFA in Hela cells [6]. In contrast, the intracellular vesicular transport of Cx26 was prevented by the depolymerization of microtubules using nocodazole, and Cx26 remained in the cell without translocating to the cell membrane to form GJPs [6]. Defourny et al. [101] came to similar conclusions using Corti organelle explants isolated from neonatal mice.

The fourth issue regards the fusion site of Cx26 with the cell membrane. Proximity ligation assays of basement membrane cultures showed that nonfunctional Cx26/30 hemichannels are present in lipid rafts, but the formed functional Cx26/30 GJPs do not colocalize with these rafts [102], and thus it was proposed that Cx26 may target lipid rafts along microtubules in the form of vesicular transport, and then undergo directional movement to form GJPs. It has also been suggested that Cx26 transport vesicles target GJPs directly along microtubules to the edges of non-lipid rafts in an actin-independent manner [101]. Thomas et al. [109] raised the possibility that Cx26 fusion with the cell membrane is Golgi-dependent, but not microtubule-dependent; that is, the Golgi extends a long tubular extension to randomly transport Cx26 to the cell membrane surface and Cx26 then undergoes lateral movement directed to the edge of GJPs. Since Qu et al. [98] showed that Cx26/30 GJPs exhibit sharp edges without the presence of perinexus regions and the assembly and stability of GJPs do not require interaction with the submembrane actin network, it is speculated that Cx26 may fuse with the cell membrane through the ER or other structures directly transported from the cell’s interior to GJPs without passing through the surrounding nonfunctional regions.

## 4. Molecules Related to Connexin Transport and Their Roles

### 4.1. Cx43-Targeted Transport

EB1 is one of the plus-end tracking proteins (+TIPs), the main role of which is to facilitate microtubule anchoring to the cell membrane [110]. The EB1 protein plays a significant role in the targeted transport of Cx43 [91]. EB1 has a dual binding site with one end directly binding to the microtubule orthotope and the other end binding to the p150(Glued), a subunit of dynactin, to form the dynein/dynactin complex [111,112,113], which then combines with β-catenin [114] to tether microtubules to AJs [115,116], in turn facilitating hemichannel transport to the cell membrane (Figure 1). N-cadherin mediates homologous interactions between adjacent cells, while β-catenin acts as a cytoplasmic enforcer of N-cadherin/N-cadherin interactions. Shaw et al. [91] showed that microtubules bound to EB1 grow more frequently and more stably toward GJPs and that GJP formation was significantly reduced after EB1 knockdown, demonstrating that EB1 at microtubule termini may interact with proteins at the cell boundary to enhance the targeted delivery of Cx43. As for the impact of N-cadherin on Cx43 transport, it was confirmed that the pre-fusion pathway of vesicles at the cell membrane in contact with the extracellular domain of N-cadherin is shorter and the incidence of fusion is higher [91]. Treatment of Hela cells with a peptide that blocks N-cadherin homologous interactions [117] resulted in a significant reduction of Cx43 GJPs at the cell-cell boundary, even without altering the location and distribution of N-cadherin [91]. This indicates the important role of N-cadherin-mediated homologous interactions between adjacent cells in the process of Cx43-directed transport and membrane fusion. Laser confocal microscopy showed that p150(Glued) is predominantly present in the cytoplasmic pool and at the cell-cell boundary close to GJPs [91]. The knockout of p150(Glued) results in a significant reduction of Cx43 on the membrane and a decrease in EB1 residence time at the cell boundary, but does not prevent the formation of AJs [91]. Similarly, there is significant colocalization of β-catenin with Cx43-YFP, and Cx43 GJPs are significantly reduced after knocking out β-catenin [91]. Studies using ventricular myocytes confirm that p150(Glued), β-catenin, and Cx43 are endogenously present in ventricular myocytes and that all three are colocalized in IDs. In Hela cells, p150(Glued) connects cadherin to the positive end of microtubules that transport Cx43. The normal distribution of AJs is also strongly associated with the transport of Cx43 and disruption of cadherin-mediated AJ formation results in a significant decrease in GJPs on the cell membrane [91]. These data suggest that AJs are a preferential transport site for Cx43 and, together, the interaction of EB1, p150(Glued), N-cadherin, and β-catenin plays an important role in the transport of Cx43.

ZO-1/2 is also involved in the transport of Cx43. The CT of Cx43 contains a PDZ-binding domain in the last three residues (V/L)-X-(V/I) [88,118]. The PDZ domain of ZO-1/2 can bind to the PDZ-binding domain of Cx43 (Figure 1) to mediate the interaction of the hemichannel with cortical actin, which in turn facilitates Cx43 hemichannel transport [23,88,98,107,119].

Eph receptors and ephrin ligands are membrane-bound cell-cell communication molecules that regulate many developmental processes by restricting cell intermingling and establishing developmental boundaries, as well as regulating the physiological functions of many developing mature organs [120,121,122]. Familial deletion of chromosome 13q33, which encodes ephrin-B2, causes comprehensive neurodevelopmental disorders including developmental delay, mental retardation, seizures, and sensorineural deafness [123]. Eph/ephrin affects intercellular gap junction communication (GJC), which is inhibited at the Eph/ephrin interface, but promoted at the ephrin/ephrin interface [122,124]. Cx43 is widely present at the interface between ephrin-B1-positive cells and in cultures of mesenchymal cells isolated from ephrin-B1^+/−^ heterozygous embryos, Cx43 was abnormally distributed and failed to form GJPs, but the overall level of Cx43 was unchanged [124], suggesting that ephrin-B1 may affect the intracellular transport of Cx43, which in turn affects the distribution of Cx43 and the formation of GJPs.

### 4.2. Cx30-Targeted Transport

Although actin plays an important role in the targeted transport of Cx30, Cx30 does not have a PDZ-binding domain (Figure 2), in contrast to the Cxs with a long CT [98]. Cx30 and ZO-1 have different distribution patterns, in which Cx30 is distributed in cell contact areas opposite to the nucleus, forming large patches, while ZO-1 is widely distributed in all cell contact areas, forming small dot-like patches [98]. Immunoprecipitation assays also showed that Cx30 does not co-precipitate with ZO-1 [98]. Therefore, Cx30 directly interacts with β-actin rather than through ZO-1/2.

Similar to ephrin-B1-mediated regulation of Cx43 transport, ephrin-B2 is associated with Cx30 transport and GJP formation [125]. By observing cochlear basement membrane cultures, it was found that Cx30 GJPs colocalize with ephrin-B2 in caveolae membrane domains. Proximity ligation assays confirm that ephrin-B2 preferentially interacts with Cx30 at the periphery of GJPs, where the newly synthesized hemichannel enters the GJP, and that the effect is transient [125] as shown in Figure 2, suggesting that ephrin-B2 may play an important role in the oriented movement of Cx30 on the membrane rather than in maintaining the stability of GJPs. Moreover, both ephrin-B2 haploinsufficiency and activation of reverse signaling increase clathrin-mediated endocytosis of Cx30 [125], which leads to a decrease in the length of the GJPs. To summarize, ephrin-B2 has an essential role in regulating Cx30 transport and degradation, but the exact molecular mechanism remains to be further explored.

### 4.3. Cx26-Targeted Transport

The literature shows that cadherin plays an important role in the targeted transport of Cx26. The main role of N-cadherin is to regulate microtubule dynamics and anchor microtubules required for targeted Cx26 vesicle transport at the cell-cell boundary (Figure 3), which in turn promotes the fusion of Cx26 transport vesicles with the cell membrane at specific sites. After treatment with GC-4, an N-cadherin neutralizing antibody, microtubules are not anchored to the cell membrane, the length of Cx26/30 GJPs is reduced, and Cx26/30 heterodimeric oligomers accumulate in the cytoplasm [102]. Disruption of microtubules using nocodazole leads to similar results. However, N-cadherin may have a less important role in the lateral movement of Cx26 across the membrane and in maintaining the stability of GJPs since the formed functional GJPs do not colocalize with either N-cadherin or N-cadherin-associated microtubules [102], as shown in Figure 3.

The +TIP EB1 is responsible for the targeted transport of Cx43 by binding to the p150(Glued) subunit of dynactin. Shaw et al. [91] suggested that EB1 may also be involved in Cx26-targeted transport. After knocking down EB1 in Hela cells, Cx26 on the cell membrane is significantly reduced and Cx26 intracellularly accumulates. This may be due to the inability of microtubules to anchor to specific regions of the cell membrane after EB1 knockdown, thereby causing Cx26 to remain in the cell and not be transported to the cell membrane.

Unlike Cx43, for which transport requires binding to actin mediated by ZO-1/2, Cx26 is not predicted to contain a PDZ-binding motif [107], so it is presumed to be unable to bind to the PDZ domain of ZO-1/2. Cx26 does not colocalize with either actin or ZO-1/2 [101]. However, whether Cx26 directly binds to actin, similar to Cx30, and thus if its targeted transport is mediated by actin, is yet to be confirmed.

Caveolins act as scaffolding proteins that aggregate lipids and signaling molecules within caveolae and also regulate the transport of proteins targeted to caveolins [126,127]. Schubert et al. [105] showed that Cx26 was effectively targeted to lipid rafts only when Cx26 and Caveolin-1 were expressed in combination, whereas Cx26 alone was excluded from lipid rafts. This suggests that Cx26 may require Caveolin-1 to target lipid rafts via a “piggyback” mechanism. In contrast, the other connexins (Cx43, Cx32, Cx36, and Cx46) are not dependent on Caveolin-1. 

Contrary to the apparent colocalization of ephrin-B2 with Cx30 GJPs, there is no colocalization of Cx26/30 GJPs with ephrin-B2 in the inner sulcus cells [125]. However, ephrin-B2 is specifically expressed in cochlear support cells [128], the ephrin/ephrin on the cell surface can promote GJC [124], ephrin can regulate the transport and degradation pathways of Cx30 [125] and Cx43 [124], and ephrin-B2 contains a PDZ-binding motif that can form a multiprotein complex with other molecules containing the PDZ domain [129] to participate in the endolymphatic potassium cycle [129,130]. Further exploration of whether ephrin-B2 is related to the transport of Cx26 is required. Molecules involved in trafficking pathways of Cxs are listed in Table 2.

## 5. Conclusions

Specific type of mutations can cause abnormalities at any particular step in the life cycle of connexins [27], result in connexins possessing a faulty subcellular localization, failing to transport to the cell membrane and preventing gap junction formation, ultimately leading to connexin dysfunction and hearing loss. Several mechanisms may be involved in transport disorders leading to connexin dysfunction and hearing loss, that is, impaired cochlear potassium recirculation [27,131], endothelial barrier breakage [132], defective gap junction facilitation of metabolite transport [133], disrupted adenosine-triphosphate-calcium signaling propagation [134,135] and dysfunctional energy supply [136].

Possible reasons for the contradictions in the literature include the following. First, the kinds of cells selected for the study are different and the biosynthetic pathways of connexins may differ between cell types. Second, there may be more than one biosynthetic pathway of Cxs, and alternative pathways may exist when the expression of Cxs increases. Third, most of the existing studies were conducted by disrupting microtubules, Golgi apparatus, and actin, followed by observing whether the Cxs could form GJPs on the cell membrane to determine whether they were associated with Cx transport. However, there are structural and functional interactions among microtubules, Golgi apparatus, and actin [137] and so disruption of one of these structures will inevitably impact the function of the others. For example, disruption of the Golgi apparatus leads to impaired formation of non-centrosomal microtubules [138], which are important for directional cell membrane transport of vesicles. Whereas microtubules are important for the distribution and localization of organelles, microtubule depolymerization induces reversible fragmentation and dispersion of the Golgi apparatus [139]. Meanwhile, microtubule depolymerization also promotes the reorganization of actin stress fibers, leading to the extension of the actin toward the bicellular junctions [140].

## 6. Perspectives

Despite abundant studies proposing models of Cx transport, there are still questions to be resolved due to technical limitations such as the temporal and spatial resolution of microscopy and phototoxicity. For example, which organelles are involved in the transport of Cxs? What are the sites where Cxs are fused to the cell membrane, and are they directly targeted to specific sites in the plasma membrane or are they randomly intracellularly transported followed by directional movement across the membrane? What molecules regulate the transport of Cxs? 

Many pathogenic mutations in Cxs not only affect the function of the channel itself but also the assembly and transport of Cxs [18,61], thus affecting the formation of GJPs. Therefore, deciphering the process of gap junction biogenesis is important to understand the pathogenic principles of these mutations and to find therapeutic strategies for hereditary hearing loss.

## Figures and Tables

**Figure 1 ijms-24-10349-f001:**
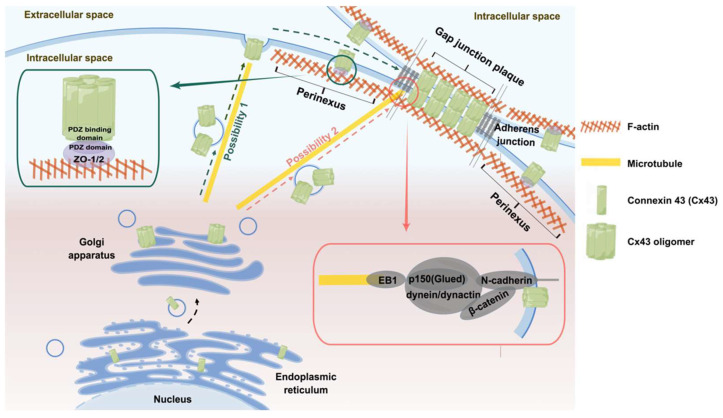
Cx43 transportation model assumptions, existing controversies, and the molecules involved in Cx43 transport and their functions. Possibility 1: The hemichannel is synthesized in the trans-Golgi, transported along microtubules to the perinexus region, and fused to the plasma membrane of the perinexus region. Then, hemichannels interact with cortical actin via ZO-1/2 [15,28], which regulates the directional transport of hemichannels from the perinexus region to the GJPs. Possibility 2: The hemichannels are synthesized in the trans-Golgi and directly transported along microtubules to GJPs. The solid arrows indicate the details of the shown sections. And the dashed arrows indicate the transport process of Cx43. (By Figdraw).

**Figure 2 ijms-24-10349-f002:**
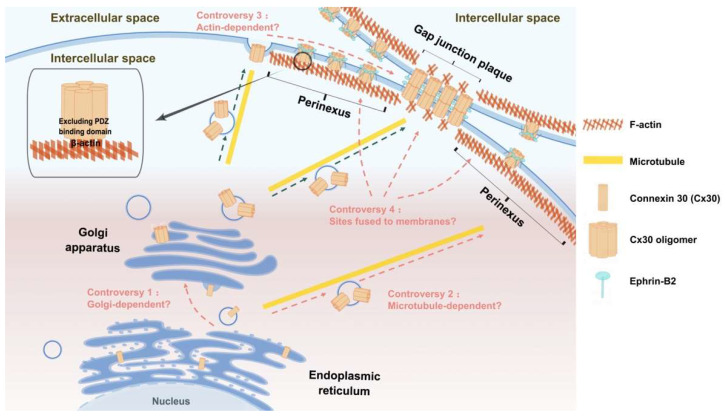
Cx30 transportation model assumptions, existing controversies, and the molecules involved in Cx30 transport and their functions. Cx30 hemichannels depend or do not depend on the Golgi apparatus and transport through microtubules to the non-GJPs region of the cell membrane, and then Cx30 hemichannels interact with actin, which regulates hemichannel transport from the non-GJPs region to the pre-existing surrounding GJPs. There are four main controversies regarding the assumptions of the Cx30 transport model, which is whether the transport of Cx30 is dependent on Golgi, microtubules, actin, and the sites of Cx30 fusion with the cell membrane. The solid arrows indicate the details of the shown sections. The dashed arrows indicate the transport process of Cx30. And the curved dashed arrows indicate the existing controversies regarding the Cx30 trafficking pathway. (By Figdraw).

**Figure 3 ijms-24-10349-f003:**
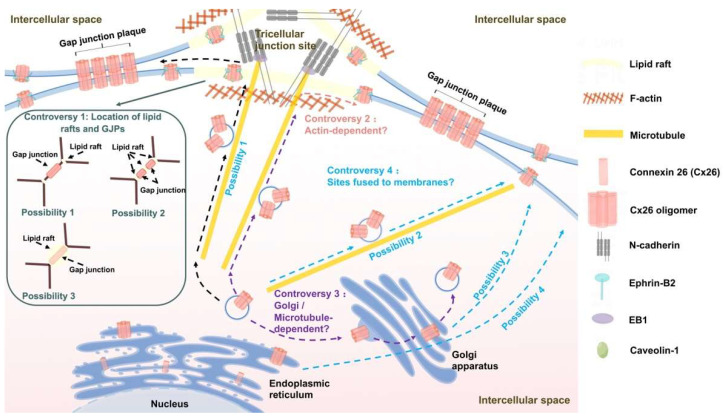
Cx26 transportation model assumptions, existing controversies, and the molecules involved in Cx26 transport and their functions. N-cadherin promotes microtubule to anchor to lipid raft structures, and Cx26/30 oligomers are transported along microtubules to the cell surface of lipid raft-rich regions. Then, due to the low affinity of Cx26 for cholesterol, Cx26/30 hemichannels targeted to lipid rafts laterally diffuse to accumulate in the periphery of preexisting GJPs. Finally, each hemichannel docks with an adjacent cellular hemichannel to form a functional gap junction. There are four controversial assumptions about the Cx26 transport model as follows. The first is the positional relationship between lipid rafts and GJPs. The second is whether Cx26 transport is dependent on actin. The third is whether the Golgi apparatus and microtubules are involved in Cx26 transport. The fourth is the site of fusion of Cx26 transport vesicles with the cell membrane; that is, by microtubule targeting to lipid rafts, or by microtubule targeting to the periphery of GJPs, or by Golgi apparatus random transport to the cell membrane surface, or by ER random transport to the cell membrane surface. The solid arrows indicate the details of the shown sections. The dashed arrows indicate the transport process of Cx26. The purple dashed arrows indicate the controversy about whether Cx26 transport is Golgi-dependent or microtubule-dependent. And the blue dashed arrows indicate the four possible bind sites of Cx26 to the cell membrane. (By Figdraw).

**Table 1 ijms-24-10349-t001:** Cx26, 30, and 43 Variants Belonging to Class I Based on in Vitro Studies.

Connexins	Variants of Connexins	References
Cx26	p.Met1Val, p.Asn14Asp, p.Gly12Valfs*2, p.Gly12Valfs*2, p.Ile20Thr, p.Ile35Ser, p.Glu42del, p.Asp50Tyr, p.Asp50Asn, p.Thr55Asn, p.Gly59Ala, p.Gly59Val, p.Cys64Ser, p.Asp66His, p.His73Arg, p.Ile82Met, p.Leu90Pro, p.Tyr136X, p.Val153Ile, p.Met163Val, p.Leu56Argfs*26, p.Pro173Arg, p.Asp179Asn, p.Arg184Pro, p.Leu214Pro, p.Leu79Cysfs*3, p.Glu147Lys, p.Phe142Leu, p.Ala88Val, p.Ala40Val, p.Asn14Lys, p.Thr86Ala, p.Ala40Gly, p.Arg32His, p.Ser199Phe, p.Phe191Serfs*5, p.Cys211Leufs*5, p.Tyr155 X, p.Trp172Cys, p.Arg184Gln, p.Gly45Glu, p.Thr86Arg, p.Ile30Asn, p.Gly12Arg, p.Lys188Arg, p.Phe191Leu, p.Val198Met, p.Gly200Arg, p.Gly200Arg, p.Ile203Lys, p.Leu205Pro, p.Thr208Pro, p.Gly59Alafs*18	[29,30,31,32,33,34,35,36,37,38,39,40,41,42,43,44,45,46,47,48,49,50,51,52,53,54,55,56,57,58,59,60]
Cx30	p.Val37Glu, p.Gly59Arg, p.Ala88Val, p.Gly11Arg, p.Ala40Val, p.Asn54Lys	[61,62,63,64,65]
Cx43	p.Ala44Val, p.Glu227Asp, p.Gly60Ser, p.Phe199Leu, p.Arg202Glu, p.Glu205Arg, p.Phe268Ala, p.Tyr155His, p.Tyr17Ser, p.Gly21Arg, p.Ala40Val, p.Leu90Val, p.Ile130Thr, p.His194Pro, p.Ile31Met, p.Leu7Val, p.Gly138Arg, p.Arg33X, p.Met147Thr, p.Arg148Gln, p.Thr154Ala, p.Arg202His	[66,67,68,69,70,71,72,73,74,75,76,77,78]

* = Termination codon.

**Table 2 ijms-24-10349-t002:** Molecules involved in connexin transport and their functions.

Connexins	Molecules Involved in Transport	Molecular Function
Cx43	EB1	+TIPs, promoting microtubule anchoring to the cell membrane.
	p150 (Glued)	Subunit of dynactin, binding to EB1 to form dynein/dunactin complexes.
	N-cadherin	Promoting microtubule tethering to AJs, and mediating homologous interactions between adjacent cells.
	β-catenin	Tethering microtubules to AJs, and acting as cytoplasmic actuators of N-cadherin/N-cadherin interactions.
	ZO-1/2	Binding to the PDZ-binding motif of connexins through the PDZ domain mediates the interaction of the hemichannel with cortical actin.
	ephrin-B1	Affecting intercellular GJC, which in turn affects the distribution of connexins and the formation of GJPs
Cx30	β-actin	β-actin directly interacts with Cx30 without via ZO-1/2.
	ephrin-B2	Preferentially interfacing with peripheral Cx30, promoting the directional movement of Cx30 across the cell membrane. Regulating clathrin-mediated Cx30 endocytosis.
Cx26	N-cadherin	Regulating microtubule dynamics, and anchoring microtubules to the cell-cell boundary.
	EB1	+TIPs, promoting microtubule anchoring to the cell membrane.
	Caveolin-1	Facilitating targeted transport of connexins to lipid rafts through a “piggy-back” mechanism.
	ephrin-B2	Whether it is involved in the transport of connexins remains to be further investigated.

## Data Availability

Not applicable.

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
