# Peer review of "Cytomembrane Trafficking Pathways of Connexin 26, 30, and 43"

_ijms, 2023, doi:10.3390/ijms241210349_

Round 1

Reviewer 1 Report

Review

to manuscript “Cytomembrane Trafficking Pathways of Connexin 26, 30 and 43: Recent updates” by Yan-Jun Zong, Xiao-Zhou Liu, Lei Tu, and Yu Sun

This review focused on discussion of the 22 transport models for connexin 43, connexins 30 and 26, mutations affecting trafficking pathways of 23 connexin 26, the existing controversies in trafficking pathways of connexins, and the molecules in- 24 volved in connexin trafficking and their functions.

In overall, this is a good job. However, I have some minor comments and recommendations.

Minor comments

1) Nomenclature of Cx26 variants presented in table 1 need for correction, according to HGVS nomenclature rules. For amino acid changes please use the one stile for example: p.Trp172Cys. For framfshift variants, please use Mutalyzer software (http://mutalyzer.nl) to present correct nomenclature. For instance, c.35delG at the protein level becomes p.Gly12Valfs*2.

2) I think the section of the summary and perspective a little so overloaded. It would be better to divide this section into two sections: For example, “Conclusion” and “Perspective” section.

Recommendation

Accept after minor comments

Author Response

To Reviewer 1

Thanks for you comments and advices. Based on your suggestions, we have made the modification to our manuscript. The following is our point to point reply.

Comment 1: Nomenclature of Cx26 variants presented in table 1 need for correction, according to HGVS nomenclature rules. For amino acid changes please use the one stile for example: p.Trp172Cys. For framfshift variants, please use Mutalyzer software (http://mutalyzer.nl) to present correct nomenclature. For instance, c.35delG at the protein level becomes p.Gly12Valfs*2.

Reply: Thanks for your comments. Based on your suggestion, we have correctted the nomenclature for the connexin variant. We have also added the variants of Cx30 and Cx43. The following is the modified Table 1:

Connexins

Variants of connexins

References

Cx26

p.Met1Val, p.Asn14Asp, p.Gly12Valfs*2, p.Gly12Valfs*2, p.Ile20Thr, p.Ile35Ser, p.Glu42del, p.Asp50Tyr, p.Asp50Asn, p.Thr55Asn, p.Gly59Ala, p.Gly59Val, p.Cys64Ser, p.Asp66His, p.His73Arg, p.Ile82Met, p.Leu90Pro, p.Tyr136X, p.Val153Ile, p.Met163Val, p.Leu56Argfs*26, p.Pro173Arg, p.Asp179Asn, p.Arg184Pro, p.Leu214Pro, p.Leu79Cysfs*3, p.Glu147Lys, p.Phe142Leu, p.Ala88Val, p.Ala40Val, p.Asn14Lys, p.Thr86Ala, p.Ala40Gly, p.Arg32His, p.Ser199Phe, p.Phe191Serfs*5, p.Cys211Leufs*5, p.Tyr155 X, p.Trp172Cys, p.Arg184Gln, p.Gly45Glu, p.Thr86Arg, p.Ile30Asn, p.Gly12Arg, p.Lys188Arg, p.Phe191Leu, p.Val198Met, p.Gly200Arg, p.Gly200Arg, p.Ile203Lys, p.Leu205Pro, p.Thr208Pro, p.Gly59Alafs*18

([29], [30], [31], [32], [33], [34], [35], [36], [37], [38], [39], [40], [41], [42], [42], [43], [44], [45], [46], [47], [48], [49], [50], [51], [52], [53], [54], [55], [56], [57], [58], [59], [60])

Cx30

p.Val37Glu, p.Gly59Arg, p.Ala88Val, p.Gly11Arg, p.Ala40Val, p.Asn54Lys

([61], [62], [63], [64], [65])

Cx43

p.Ala44Val, p.Glu227Asp, p.Gly60Ser, p.Phe199Leu, p.Arg202Glu, p.Glu205Arg, p.Phe268Ala, p.Tyr155His, p.Tyr17Ser, p.Gly21Arg, p.Ala40Val, p.Leu90Val, p.Ile130Thr, p.His194Pro, p.Ile31Met, p.Leu7Val, p.Gly138Arg, p.Arg33X, p.Met147Thr, p.Arg148Gln, p.Thr154Ala, p.Arg202His

([66], [67], [68], [69], [70], [70], [71], [72], [73], [74], [75], [76], [77], [78])

Comment 2: I think the section of the summary and perspective a little so overloaded. It would be better to divide this section into two sections: For example, “Conclusion” and “Perspective” section.

Reply: Thanks for your comments. We have deleted many paragraphs and sentences to make this section more concise. In addition, we have divided this section into two sections. The following is the revised “Conclusion” and “Perspective” section.

  1. Conclusions

Specific type of mutations can cause abnormalities at any particular step in the life cycle of connexins [27], result in connexins possessing a faulty subcellular localisation, failing to transport to the cell membrane and preventing gap junction formation, ulti-mately leading to connexin dysfunction and hearing loss. Several mechanisms may be involved in transport disorders leading to connexin dysfunction and hearing loss, that is, impaired cochlear potassium recirculation [27, 131], endothelial barrier breakage [132], defective gap junction facilitation of metabolite transport [133], disrupted adeno-sine-triphosphate-calcium signaling propagation [134, 135] and dysfunctional energy supply [136].

Possible reasons for the contradictions in the literature include the following. First, the kinds of cells selected for the study are different and the biosynthetic pathways of connexins may differ between cell types. Second, there may be more than one biosynthetic pathway of Cxs, and alternative pathways may exist when the expression of Cxs increases. Third, most of the existing studies were conducted by disrupting microtubules, Golgi ap-paratus, and actin, followed by observing whether the Cxs could form GJPs on the cell membrane to determine whether they were associated with Cx transport. However, there are structural and functional interactions among microtubules, Golgi apparatus, and ac-tin [137] and so disruption of one of these structures will inevitably impact the function of the others. For example, disruption of the Golgi apparatus leads to impaired formation of non-centrosomal microtubules [138], which are important for directional cell membrane transport of vesicles. Whereas microtubules are important for the distribution and locali-zation of organelles, microtubule depolymerization induces reversible fragmentation and dispersion of the Golgi apparatus [139]. Meanwhile, microtubule depolymerization also promotes the reorganization of actin stress fibers, leading to the extension of the actin to-ward the bicellular junctions [140].

  1. Perspectives

Despite abundant studies proposing models of Cx transport, there are still questions to be resolved due to technical limitations such as the temporal and spatial resolution of microscopy and phototoxicity. For example, which organelles are involved in the transport of Cxs? What are the sites where Cxs are fused to the cell membrane, and are they directly targeted to specific sites in the plasma membrane or are they randomly transported intracellularly followed by directional movement across the membrane? What molecules regulate the transport of Cxs?

Many pathogenic mutations in Cxs not only affect the function of the channel itself but also the assembly and transport of Cxs [18, 61], thus affecting the formation of GJPs. Therefore, deciphering the process of gap junction biogenesis is important to understand the pathogenic principles of these mutations and to find therapeutic strategies for heredi-tary hearing loss.

Reviewer 2 Report

In their manuscript entitled “Cytomembrane Trafficking Pathways of Connexin 26, 30 and 43: Recent updates.”, Zong et al present an overview of the main pathways and controversies associated with the trafficking of these proteins and their functions. Overall, the manuscript is of general interest to a broad readership, but some clarifications are required before publication.

Hearing loss is indicated in the abstract as the main topic of this review. If so, it should be mentioned in the manuscript title in order to better describe its content. However, while the abstract seems to indicate that hereditary deafness will be central to the review, it is not mentioned in chapters 3 and 4. Some clarification regarding the precise goal of this review should be brought by the authors, and necessary changes performed before publication of this manuscript (e.g., providing examples directly related to deafness or taken from studies on cells from the inner ear whenever possible).

The title indicates that this review will focus on “recent updates”. However, a majority of the references used are more than 10 year old, making this statement untrue. Some rephrasing is thus necessary, or a better emphasis on recent publications.

Chapter 2 lists the main Cx26 mutations associated with deafness in human patients. There is however no similar chapter for Cx30 and Cx43. Are we to understand that only mutations in Cx26 are linked with this disease? The manuscript should describe Cx30/43 mutations, or justify why they are not relevant here.

Moderate editing of English language required

Author Response

To Reviewer 2

Thanks for you comments and advices. We have made the modification to our manuscript according to your advices. The following is our point to point reply.

Comment 1: Hearing loss is indicated in the abstract as the main topic of this review. If so, it should be mentioned in the manuscript title in order to better describe its content. However, while the abstract seems to indicate that hereditary deafness will be central to the review, it is not mentioned in chapters 3 and 4. Some clarification regarding the precise goal of this review should be brought by the authors, and necessary changes performed before publication of this manuscript (e.g., providing examples directly related to deafness or taken from studies on cells from the inner ear whenever possible).

Reply: Thanks for your comments. This review aims to summarize and describe the existing research findings of cytomerane traffic pathways of connexin 26, 30, and 43. As connexin must traffic to the cytomerane of adjacent cells and form gap junctions to function normally, variants of genes encoding connexin may cause disruptions in the trafficking pathways and subcellular localization of mutated connexin proteins. Therefore, our article mainly focuses on mutations that may affect the trafficking pathways of connexin, possible trafficking pathways of connexin, and molecules that may play a key role in the cytomerane trafficking pathways of connexin. We truely mentioned in the abstract that "This review can contribute to a new way of understanding the etiological principles of connexin mutations and finding therapeutic strategies for hereditary deafness." The original intention is only to highlight the value of this study for clinical doctors and researchers in related fields. Hearing loss is not the main theme of this manuscript, and we apologize for any misunderstandings caused by these words. Therefore, we have revised the last sentence of the abstract. We have change the sentence “This review can contribute to a new way of understanding the etiological principles of connexin mutations and finding therapeutic strategies for hereditary deafness” to “This review can contribute to a new way of understanding the etiological principles of connexin mutations”. In addition, we have added a description of the possible mechanism of connexin causing hearing loss in line 115 of this manuscript. The following is the added sentences:

The material exchange and information exchange between adjacent cells are important functions of connexin. Part of the mutations listed in Table 1 have been confirmed to affect the permeability changes of cells to ions or other molecules in vitro. Therefore, the abnormal traffic pathways and subcellular localization of mutated connexin may be important for the pathological processes of diseases such as hearing loss.

Comment 2: The title indicates that this review will focus on “recent updates”. However, a majority of the references used are more than 10 year old, making this statement untrue. Some rephrasing is thus necessary, or a better emphasis on recent publications.

Reply: Thanks for your comments. We have removed “recent updates” from the title. In fact, there has been no breakthrough in the research field on cytomerane traffic pathways of connexin in recent years. Perhaps higher resolution microscopes can help research in this field.

Comment 3:Chapter 2 lists the main Cx26 mutations associated with deafness in human patients. There is however no similar chapter for Cx30 and Cx43. Are we to understand that only mutations in Cx26 are linked with this disease? The manuscript should describe Cx30/43 mutations, or justify why they are not relevant here.

Reply: Thanks for your comments. We have summarized and added the the mutations of Cx30 or Cx43 that may cause disruption in the trafficking pathways and subcellular localization. The following is the modified Table 1:

Connexins

Variants of connexins

References

Cx26

p.Met1Val, p.Asn14Asp, p.Gly12Valfs*2, p.Gly12Valfs*2, p.Ile20Thr, p.Ile35Ser, p.Glu42del, p.Asp50Tyr, p.Asp50Asn, p.Thr55Asn, p.Gly59Ala, p.Gly59Val, p.Cys64Ser, p.Asp66His, p.His73Arg, p.Ile82Met, p.Leu90Pro, p.Tyr136X, p.Val153Ile, p.Met163Val, p.Leu56Argfs*26, p.Pro173Arg, p.Asp179Asn, p.Arg184Pro, p.Leu214Pro, p.Leu79Cysfs*3, p.Glu147Lys, p.Phe142Leu, p.Ala88Val, p.Ala40Val, p.Asn14Lys, p.Thr86Ala, p.Ala40Gly, p.Arg32His, p.Ser199Phe, p.Phe191Serfs*5, p.Cys211Leufs*5, p.Tyr155 X, p.Trp172Cys, p.Arg184Gln, p.Gly45Glu, p.Thr86Arg, p.Ile30Asn, p.Gly12Arg, p.Lys188Arg, p.Phe191Leu, p.Val198Met, p.Gly200Arg, p.Gly200Arg, p.Ile203Lys, p.Leu205Pro, p.Thr208Pro, p.Gly59Alafs*18

([29], [30], [31], [32], [33], [34], [35], [36], [37], [38], [39], [40], [41], [42], [42], [43], [44], [45], [46], [47], [48], [49], [50], [51], [52], [53], [54], [55], [56], [57], [58], [59], [60])

Cx30

p.Val37Glu, p.Gly59Arg, p.Ala88Val, p.Gly11Arg, p.Ala40Val, p.Asn54Lys

([61], [62], [63], [64], [65])

Cx43

p.Ala44Val, p.Glu227Asp, p.Gly60Ser, p.Phe199Leu, p.Arg202Glu, p.Glu205Arg, p.Phe268Ala, p.Tyr155His, p.Tyr17Ser, p.Gly21Arg, p.Ala40Val, p.Leu90Val, p.Ile130Thr, p.His194Pro, p.Ile31Met, p.Leu7Val, p.Gly138Arg, p.Arg33X, p.Met147Thr, p.Arg148Gln, p.Thr154Ala, p.Arg202His

([66], [67], [68], [69], [70], [70], [71], [72], [73], [74], [75], [76], [77], [78])